# Exploring the interior of 3D endoluminal lesions in the air spaces by a novel electronic biopsy technique: A preliminary study of endoluminal colon lesions

Lih-Shyang Chen[1], Shao-Jer Chen [2,3]*, Ta-Wen Hsu[2,4], Shu-Han Chang[1], Chun-Ju Hou[5], Chih-Wen Lin[2,3], Yu-Ruei Chen[2,3], Chin-Chiang Hsieh[6], Shu-Chen Han[7], Ku-Yaw Chang[8]

**1** Department of Electric Engineering, National Cheng Kung University, Tainan, Taiwan, ROC, **2** School of Medicine, Tzu Chi University, Hualien, Taiwan, ROC, **3** Department of Medical Imaging, Dalin Tzu Chi Hospital, Buddhist Tzu Chi Medical Foundation, Chiayi, Taiwan, ROC, **4** Department of General Surgery, Dalin Tzu Chi Hospital, Buddhist Tzu Chi Medical Foundation, Chiayi, Taiwan, ROC, **5** Department of Electrical Engineering, Southern Taiwan University of Science and Technology, Tainan, Taiwan, ROC, **6** Department of Radiology, Tainan Hospital, Ministry of Health and Welfare, Tainan, Taiwan, ROC, **7** Department of Radiology, Tainan Municipal Hospital, Tainan, Taiwan, ROC, **8** Department of Computer Science and Information Engineering, Da-Yeh University, Changhua, Taiwan, ROC

☯ These authors contributed equally to this work.

* shaojer.chen@msa.hinet.net

**Data Availability Statement:** Data are available from the Buddhist Dalin Tzu Chi General Hospital

## Abstract

To explore the interior of a lesion in a 3D endoluminal view, this study investigates the application of an 'electronic biopsy' (EB) technique to computed tomographic colonography (CTC) for further differentiation and 2D image correlation of endoluminal lesions in the air spaces. A retrospective study of sixty-two various endoluminal lesions from thirty patients (13 males, 17 females; age range, 31 to 90 years) was approved by our institutional review board and evaluated. The endoluminal lesions were segmented using gray-level threshold and reconstructed into isosurfaces using a marching cube algorithm. EB allows users to interactively erode and apply grey-level mapping (GM) to the surface of the region of interest (ROI) in 3D CTC. Radiologists conducted the clinical evaluation, and the resulting data were analyzed. EB significantly improves 3D gray-level presentation for evaluating the surface and inside of endoluminal lesions over that of SR, GM or target GM (TGM) (P < 0.01) with preservation of the 3D spatial effect. Moreover, 3D to 2D image correlation were achieved in any layer of the lesion using EB as did GM/TGM on the surface. The specificity and diagnostic accuracy of EB are significantly greater than those of SR (P < 0.01). These performance can be better further with GM/TGM and reach the best with EB (specificity, 89.3–92.9%; accuracy, 95.2–96.8%). EB can be used in CTC to improve the differentiation of endoluminal lesions. EB increases 3D to 2D image correlations of the lesions on or beneath the lesion surface.

Institutional Data Access / Ethics Committee (contact via 886-5-2648000 ext. 5908) for researchers who meet the criteria for access to confidential data. Please refer to the below linkage to where and how the code has been shared. https://github.com/Shao-Jer/OnSphereErosionROIMod.

**Funding:** National Science Council, NSC 99-2221-E-303 -002 -MY3, Dr. Shao-Jer Chen Dalin Tzu Chi Hospital, DTCRD 99(1)-15, Dr. Shao-Jer Chen.

**Competing interests:** The authors have declared that no competing interests exist.

## Introduction

Air and food are essential substances taken from outside to support human life. They are transported and connected by many tubular structures to vital organs. However, many pathogens are also easily carried by them through tubules into the human body and cause diseases. For example, the recent Coronavirus disease (COVID-19) pandemic is due to virus infection through the tubular airway. The human lives and societies are threatened by it greatly. Colorectal cancer is another common form of gastrointestinal tract cancer and a leading cause of cancer-related death in the world [1].

Endoluminal viewing is important for the diagnosis and treatment of intraluminal lesions in tubular organs, a finding that accompanies a variety of medical problems. For example, the preliminary diagnosis of airway disease (infection or neoplasm), colon cancer, esophageal cancer, gastric cancer, and urinary bladder cancer depend on endoscopic examination. However, traditional optical endoscopy (OE) has many limitations. First, OE cannot see through endoluminal lesions, precluding evaluation of the nature of the lesion beneath surface and its extraluminal extension. Second, OE cannot pass through regions of stenosis, such that lesions distal to the narrowed area cannot be evaluated. Third, OE is uncomfortable for the patient and poses a risk of perforation [2, 3].

Computed tomographic endoscopy (CTE) is an important computer-based alternative to traditional endoscopy. CTE complements OE by providing earlier screening, allowing the visualization of additional anatomic features hidden behind stenotic regions or beneath the lesion surface, and surgical planning.

Despite the usefulness of CTE, the traditional surface rendering (SR) technique used in this technology has two drawbacks. First, SR converts the data into a simplified, binary form for surface area estimation [4]. Since no gray-level information for the endoluminal lesion surface is visible, the specificity and diagnostic accuracy for imaging intra- and endoluminal lesions using only SR is poor [5]. Second, SR only evaluates the surface information of 3D endoluminal lesions. To facilitate the discrimination and management of such lesions, the exploration of gray-level changes below the surface of endoluminal lesions warrants further study.

Volume rendering (VR) is also widely used in CTE to distinguish between true and false polyps. On a typical monochromatic volume-rendered 3D endoluminal view, a polyp cannot be distinguished from polyp-like tagged stool without reference to translucency maps [6]. Translucency mapping requires the assignment of blue, green, red, and white color channels to areas of increasing attenuation [7], a process that is subjective and influenced by user settings.

Chen et al. report a GM technique that modifies the traditional surface rendering images by applying gray level to the surface points in CTC [5]. The technique is reported to be more accurate than the traditional surface rending method in correlating CTC and 2D views, thereby improving the identification and differentiation of endoluminal lesions. In contrast to the abstract color transfer function used in VR, the gray-level range and character mapped in Chen's technique are the same as those in the original CT images, making them easier for radiologists and physicians to interpret.

To speed up the navigation process and highlight gray-level patterns in endoluminal lesions, Chen et al. also developed a TGM technique [8]. Users can select any region of interest for further 3D analysis by this method. However, both the GM and TGM techniques only survey gray-level information from the surface points of a 3D object. Radiologists are still unable to visualize tissues below the surface in the endoluminal view to investigate a suspicious lesion.

To help users evaluate gray-level information in 3D endoluminal lesions below the surface, this study investigates an electronic biopsy method that would allow for further exploration

and differentiation of such lesions. Similar to the endoscopic biopsy, the purpose of the study is the internal exploration of 3D endoluminal lesions using an EB method. CTC cases were used as an example to demonstrate the 3D shape, texture effect, image correlation of the EB method and its impact on clinical diagnosis. This study may be extended to the endoluminal lesions in other air filled spaces, such as airway and upper gastrointestinal systems.

## Materials and methods

This retrospective preliminary study of prospectively acquired data was approved by Buddhist Dalin Tzu Chi General Hospital Institutional Review Board. Informed consent was obtained from all patients before the CT study, optic colonoscopy, surgery, and pathologic examination for suspicious colon lesions. They agreed to receive CTC for better localization of the lesions and diagnosis than the conventional abdominal CT. All methods mentioned below were carried out in accordance with relevant guidelines and regulations. The flow chart methodology of this research is shown in the Fig 1.

### Data acquisition

A spiral CT scan of the patient's abdomen was performed after the entire colon was distended with room air and a tagging agent used beforehand (200 ml Telebrix 1-day preparation) [9]. Helical CT scans of the prone and supine positions were performed 30 seconds after intravenous contrast administration with the following parameters: collimation 64×0.6 mm; gantry rotation time 0.8 s; X-ray Tube Current: 140 mA; Exposure Time: 800 ms; kVp 120. Scanning was performed craniocaudally during a single breath-hold. With the advance of multiple-slice CT, the slice thickness of volume data can be reconstructed to 1.25 mm. All scans were performed using a 64-slice GE Lightspeed VCT scanner (GE HealthCare, Milwaukee, Wisconsin, USA).

From March, 2010 to May, 2013, 62 endoluminal lesions of various types from 30 patients (13 men, 17 women; age range. 31–90 years) were recruited into the study. The lesions included 34 potential malignancies (5 stalk and 5 flat polyps, size: 6.8–29.9 mm, and 24 cancerous mass) and 28 benign conditions (stool, fluid retention, calcifications/tagging agent, and ileocecal valves). All the potential malignancies were proved by the optic colonoscopy, and pathologic examination.

### Segmentation

The range of the gray-level threshold was adjusted until the soft tissue of the abdomen and the air-distended colon could be completely separated. The above process can be done by a defined gray level threshold regardless different patients and regions because colon air is within a narrow range of the CT number centered around -1000 Hounsfield units (HUs) in theory. To generate good EB effect, air areas should be segmented and separated as much as possible.

### Finding the object surface points and SR

Marching Cube is an algorithm for detecting isosurfaces in volumetric data sets that has been used to find the surface points of the colon in many CTC applications [10]. The premise of this method is that a voxel (cube) can be defined by the pixel values at the eight corners of the cube. The surface intersects those cube edges where one vertex is outside the surface (one) and the other is inside the surface (zero). Because the cube has two different symmetries, there are 14 possible patterns to describe how a surface intersects the cube. By determining which edges of the cube are intersected by the isosurface, we can create triangular patches that divide the cube into regions within and outside the isosurface. Connecting the patches from all cubes on the isosurface boundary results in a surface representation [11].

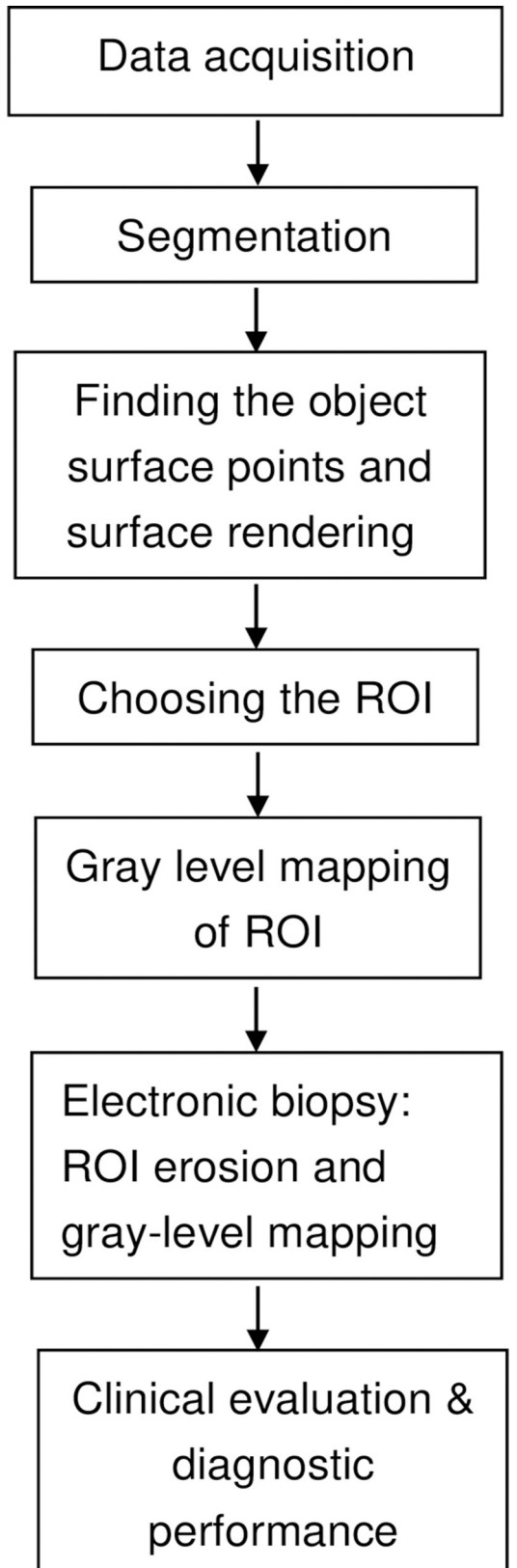

**Fig 1. Flow chart of methodology.**

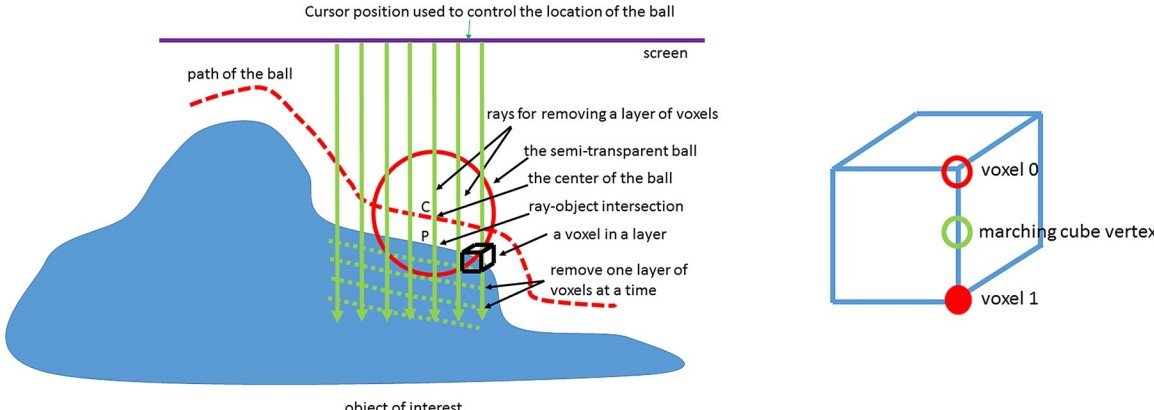

**Fig 2. Selection of surface points with GM and EB of them.** An object is created using threshold segmentation and the marching cubes algorithm. (a) A ray going through the center of the ball (C) is fired and the P, i.e., the ray-object intersection point, is computed. The center of the ball is continuously adjusted to keep the distance between C and P a constant ratio of the ball radius so that the ball can be moved along the surface of the object of interest properly. The ROI is defined by the surface points inside the range of ball along its path. Then the EB of the ROI is implemented by emitting rays from screen to the object surface at the viewing direction, removing voxels one layer at a time and forming a new surface (green dash lines). (b) A 3D grid cube containing the surface is evaluated as the black cube in (a). A marching cube triangle will intersect the cube with marching cube vertex between voxels 1 and 0 representing the inside and outside the object. In GM algorithm, the grey level value of voxel 1 is applied to marching cube vertex to construct a surface. The same application can be used in the new surface after the EB.

## Choosing the ROI

Many methods can be used in ROI setting. One way to mark the ROI on a 3D surface is to display a semi-transparent ball in front of the lumen and enable the user to use his cursor to control the location of the ball to identify the region of interest (Fig 2A). After proper setting of the radius of the ball and its sink ratio [8], the ball will go along the curvature of the lumen as the dotted red line shown in Fig 2A. The surface points (i.e., the vertices of the colon wall) inside the range of the ball along its path are gray-level mapped. If detailed setting ROI is not necessary, then simply move the mouse to a point of CTC scene and click the mouse button. The surface points inside a fixed preset area centered of the mouse position will be chosen.

## GM of ROI

To assign the gray level to the ROI surface points, the 3D grid cube on the boundary is evaluated (Fig 2A black cube). The voxel 1 and voxel 0 represent internal and external voxels of the object with 1 and 0 values respectively (Fig 2B). The marching cubes triangle formed by the marching cube vertices intersect the edges of the 3D grid (Fig 2B). To represent the internal object surface in GM algorithm, the grey level of the marching cubes vertex takes the grey level of voxel 1, the closest internal voxel. The vertices outside the ROI will remain unchanged.

## Electronic biopsy: ROI erosion and gray-level mapping

To further investigate the grey level beneath the colon surface, the user removes voxels on the colon surface one layer at a time, gradually digging into the colon wall in the direction perpendicular to the user's perspective. In other words, the user can gradually erode the colon surface inward into the colon wall along the viewing direction by controlling the number of layers of voxels removed. This procedure, known as EB, is similar to that used in physical biopsies. EB is implemented by an algorithm similar to that used for ray-tracing, which emits rays from the viewing direction toward the colon surface and removes voxels one layer at a time (Fig 2A).

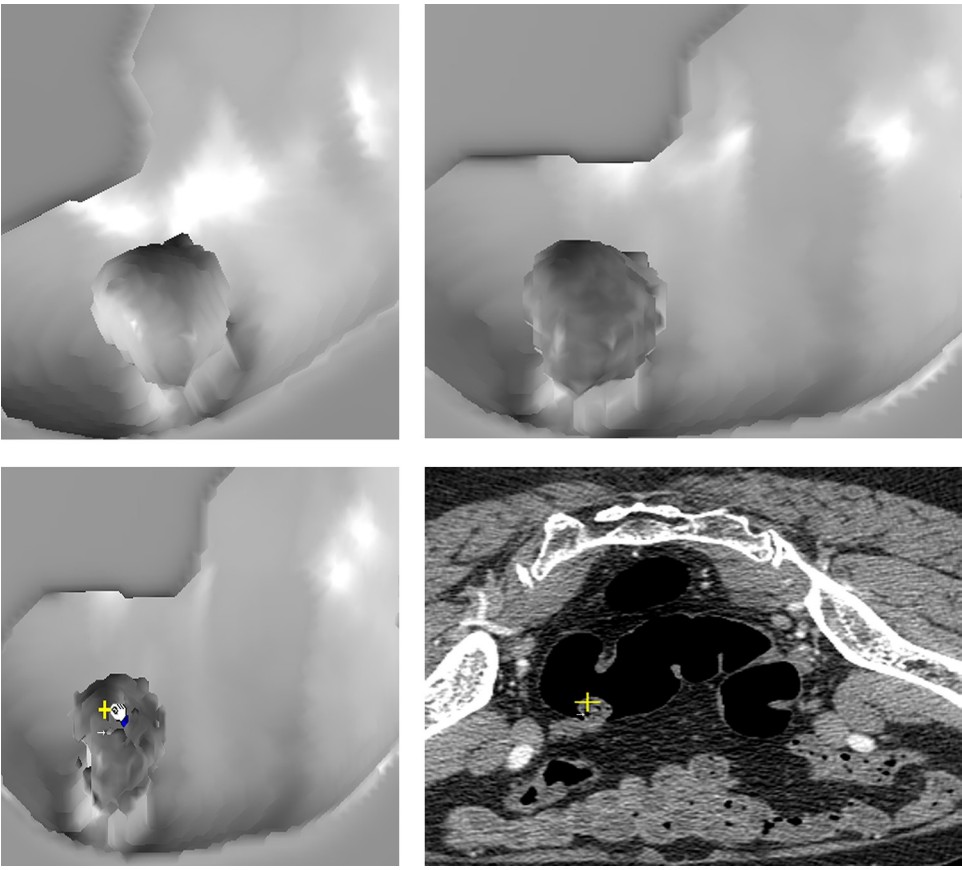

**Fig 3. A 60-year-old female with descending colon cancer diagnosed by colonoscopy.** CTC found a polypoid lesion in the sigmoid colon. (a) The polyp appears suspicious in traditional SR CTC. (b) Rough surface and heterogeneous hypo-densities of the polypoid lesion are suspicious in the initial EB image. Stool is suspicious but not a polyp. (c) EB of the polypoid lesion 4 voxels beneath the surface. Persistent and more apparent hypodensities (cross) are found in EB. Small cavitation (arrow) is also noted in EB. These findings are characteristic of stool. (d) The corresponding 2D position of the hypo-densities (cross) in EB below surface (HU, −110). The corresponding 2D position of cavitation (arrow).

After the voxels are removed, the GM algorithm is applied again to the new colon surface. Users can do this procedure over and over again into the colon wall to investigate a potential lesion as to its depth and size. Since the new colon surface covers a relatively small area, this procedure can be accomplished in real time.

## Clinical evaluation

Two radiologists with more than 10 years of experience reading abdominal CTs and specific training in CTC were invited for the retrospective evaluation. They were asked to compare the SR, GM, TGM to the electronic biopsy images and then the original CT images for each case. Both radiologists were blinded to the other's results. For EB image quality assessment, four criteria were used to evaluate the target endoluminal lesions with SR, GM, TGM and EB images: (1) spatial shape of the object surface, (2) gray-level pattern of the object or EB surface, and (3) the correlation between the 3D CTC and 2D CT and vice versa of the surface and inner presentation. The SR, GM, TGM and EB images were scored using a 3-point scale as follows: 1, poor; 2, fair; and 3, excellent. A higher score indicates better spatial surface/gray-level pattern/

correlation effect. The quantity agreement in the marginal totals between the 2 observers were also measured.

## Diagnostic performance

Another two gastrointestinal radiologists with 21 and 9 years of experience respectively evaluated the diagnostic performance of the SR, GM, TGM and EB images. The 4 types of CTC navigation images for each case were presented side-by-side to the radiologists for direct comparison. The radiologists were asked to judge the lesions as benign or potentially malignant in the SR, GM, TGM and EB CTC images alone, without cross reference to the 2D images. Both radiologists were blinded to any information of the lesions. The sensitivity, specificity and diagnostic accuracy of the above test will be analyzed.

The Wilcoxon signed ranks test was used to analyze the clinical evaluation data. All statistical analyses were performed using the Statistical Package for the Social Sciences (SPSS) statistical software for Windows, version 22 (SPSS Inc., Chicago, USA). A probability (P<0.05) is considered statistically significant.

## Results

### EB of endoluminal lesions in CTC images

Fig 3A show traditional SR image of a polypoid lesion. The initial EB image (Fig 3B) of the polypoid lesion reveals suspicious heterogeneous low density on the surface. Further EB of the lesion to 4 pixels depth beneath the surface shows evident cavitation inside, which is a characteristic finding of stool caused by surface rendering of small air spaces (Fig 3C arrow) and hypo-densities (Fig 3C cross). Fig 4A shows another example of a broad-based endoluminal lesion in an SR CTC image. The initial EB image shows mildly increased heterogeneous density (Fig 4B). Further EB up to 5 pixels depth shows the same increased density in each layer, without hypo-density or cavitation (Fig 4C). Fig 5A shows a prominent fold in the SR endoluminal view. The fat density of the fold lesion (Fig 5B arrow) is suspicious in the initial EB images rather than in the SR images. Fig 5C shows persistent fat density in the subsequent layers beneath the surface during EB (Fig 5C arrow). The 2D CT image shows the fat content of a lipomatous colon lesion (Fig 5D). The last example of a polypoid lesion in SR CTC (Fig 6A) shows a dense tagging fluid coating on the surface and inferior layering of the lesion in GM mode (Fig 6B). Subsequent EB shows the polypoid nature of the lesion, with mildly increased density below the dense surface (Fig 6C). EB methods successfully correlate the contour and gray level of any 3D point (Figs 3C–6C) in each layer inside the surface with its 2D image (the cross of Figs 3D–6D).

### Clinical evaluation

EB preserved the 3D spatial shape, gray-level pattern and images correlation of the endoluminal lesions during the biopsy procedure (Table 1) as previous described GM or TGM methods [5, 8]. EB scored significantly higher than did SR in the criteria evaluated by the two radiologists, regardless of the malignant or benign status as determined pathologically (one-tailed Wilcoxon signed ranks test, P < 0.01). The power values of the paired tests are also high (> 0.8). According to the localization and gray level correspondence, EB images provided a significantly higher correlation of 3D to 2D images than did SR images (one-tailed Wilcoxon signed ranks test, P < 0.01) with high power above 0.8 (Table 1). The classifications performed by both observers were in very good agreement—that is, better EB experience than SR (Table 1). In the evaluation of the rendering effect of the new surface after each biopsy, EB

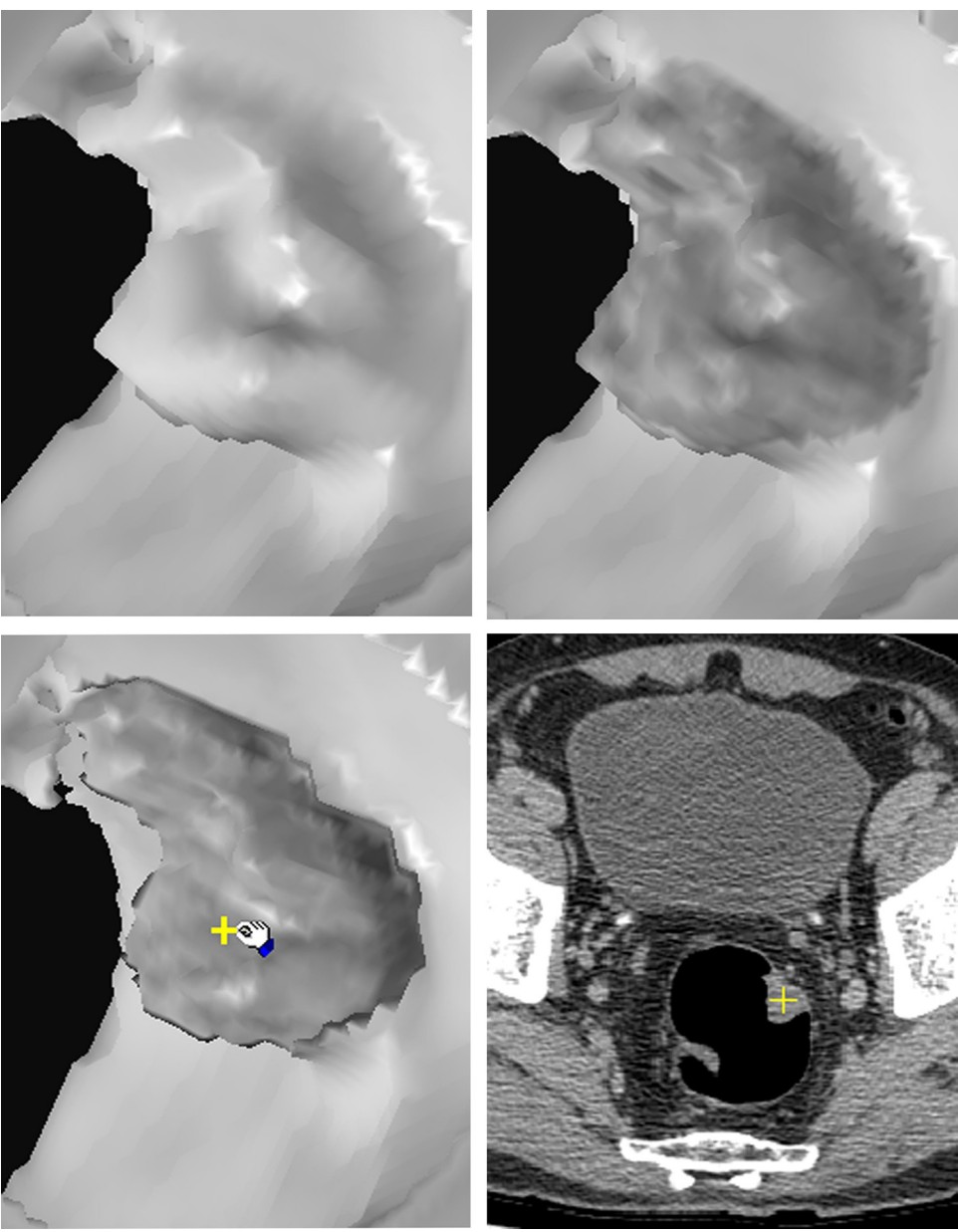

**Fig 4. CTC images of a 60-year-old male with surgically proven rectal cancer.** Comparison of SR and EB of colon cancer on CTC. (a) SR of the polypoid rectal cancer (b) Beginning of EB in the first layer of the lesion shows mildly homogeneous, increased density that is on the contrary in Fig 3(B). (c) EB of the lesion shows persistently homogeneous and increased density up to 5 pixels deep to the surface. No low-density areas or cavitation inside the lesion were found during the EB procedure. (d) The correlated 2D position of the Fig 4(c) (cross) in the EB (HU, 57).

images reveal increased gray level pattern (texture) changes with statistically difference compared with GM or TGM methods (P<0.01). No statistical difference of the spatial and correlation effect between the EB and GM/ TGM.

## Diagnostic performance

Performance studies showed EB retains high sensitivity to predict potential malignancies as GM or TGM methods (Table 2) [5, 8]. Using TGM or EB to investigate deeper layers of a

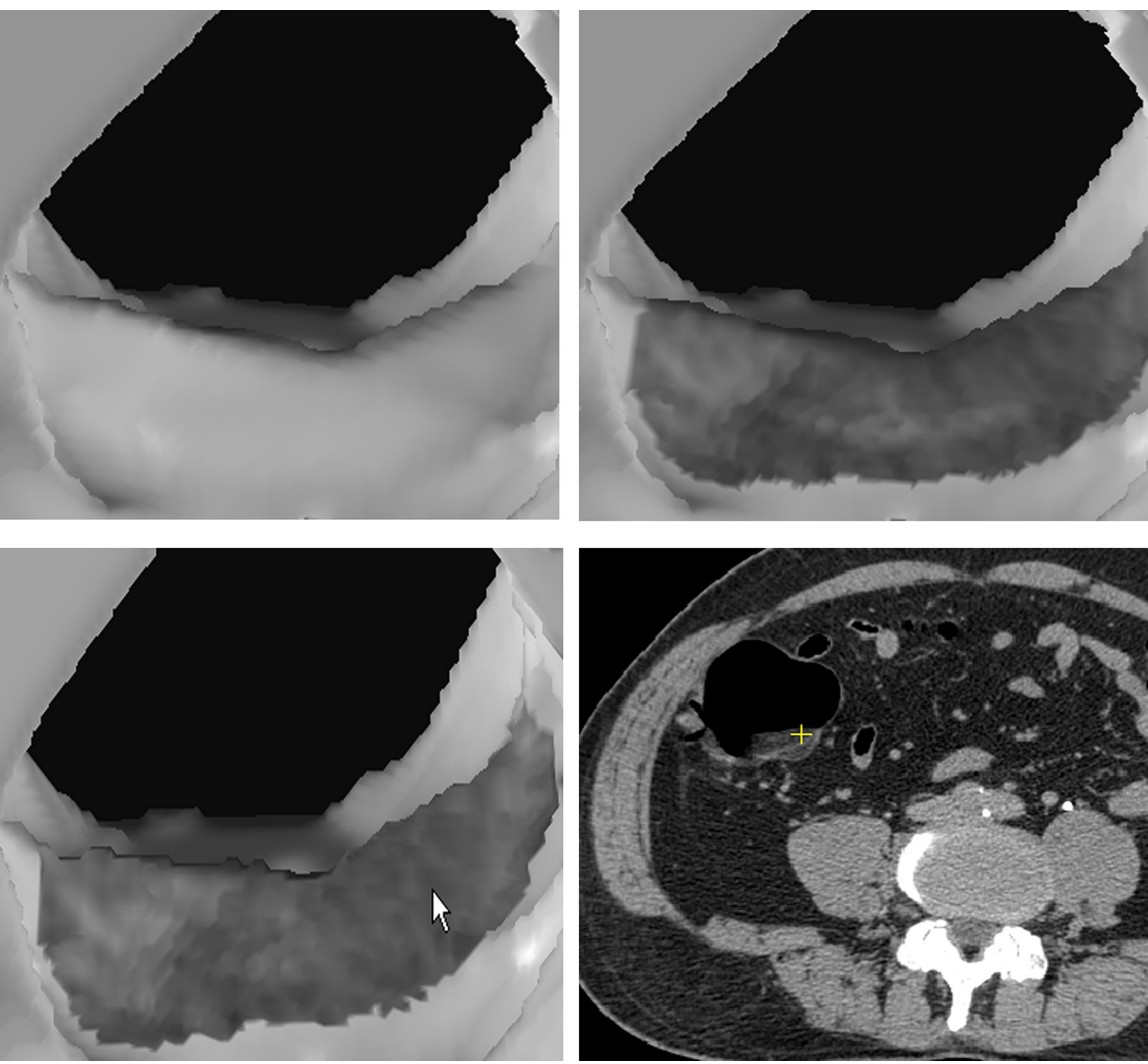

**Fig 5. Images from a 62-year-old male.** Colonography revealed swelling of the ileocecal valve with narrowing of the terminal ileum. CTC of the prominent ileocecal fold: (a) SR of the fold. (b) Beginning of EB reveals multiple hypodensities on the fold (arrow). (c) Further EB study shows the same hypodensities in the subsequent layer beneath the surface. (d) The original 2D position in c (cross). The density indicates fat (HU, −88).

lesion increased the sensitivity to 100%. The major disadvantages of SR are its low specificity of benign lesions (14.3–17.9%) and low diagnostic accuracy, leading to no significant improvement in diagnostic performance (P = 0.082). GM or TGM has known to significantly increase the specificity and diagnostic accuracy, yielding significantly improved diagnostic performance in comparison with SR [5, 8]. The results are further improved if deeper layers are imaged by EB (specificity, from 75.0–85.7% to 89.3–92.9%; diagnostic accuracy, from 88.7–93.5% to 95.2–96.8%) (Table 2).

## Discussion

We have demonstrated the EB effects on commonly seen endoluminal colon lesions via the figures in the result portion. The HU (gray level scale) and 3D spatial shape can be displayed simultaneously from endoluminal view either inside or on the surface of a lesion. The visual

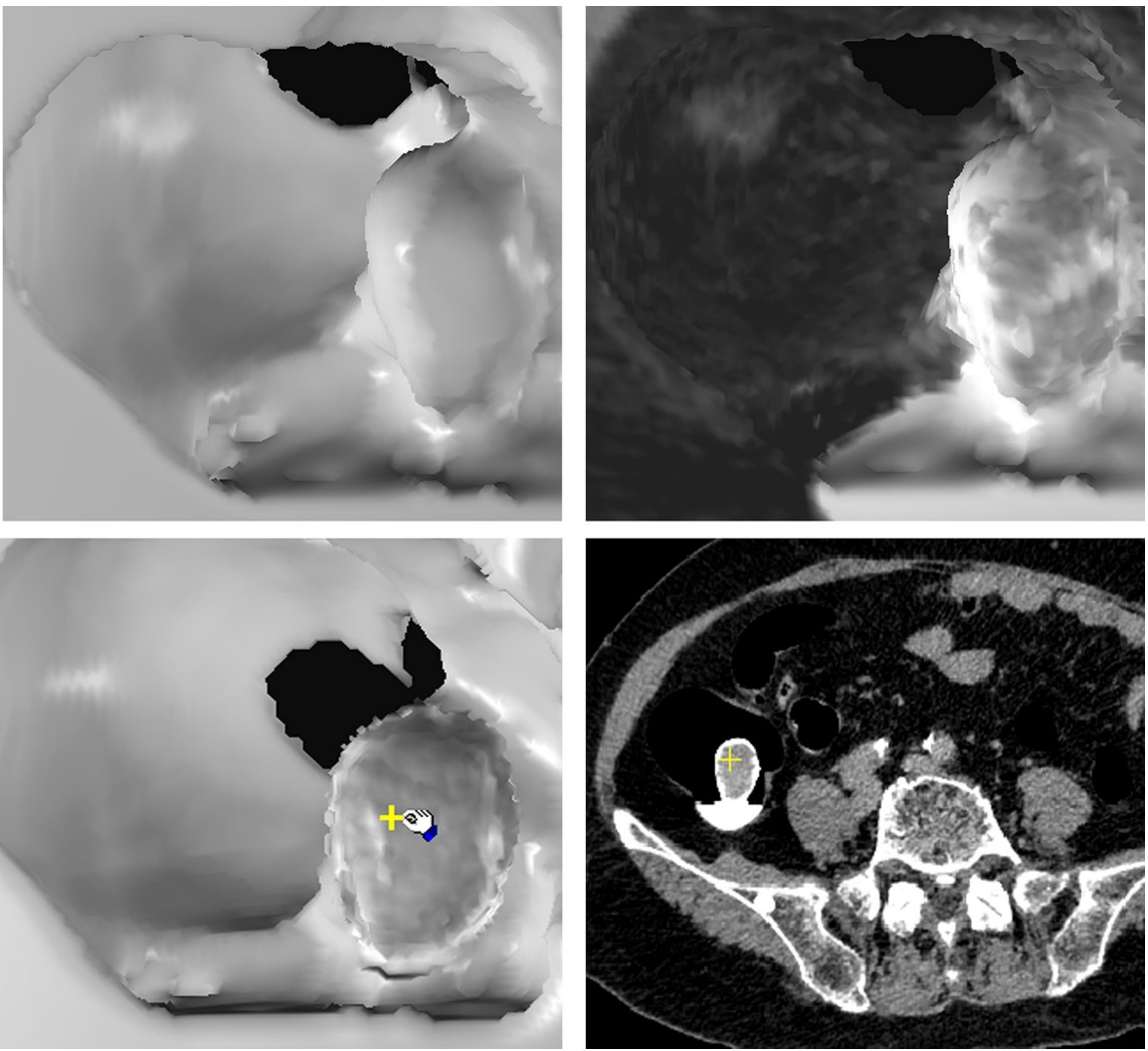

**Fig 6. CTC evaluation of an endoluminal lesion below tagging agents.** A 72-year-old female with a biopsy-proven cecal adenocarcinoma diagnosed at the local medical department. (a) SR shows only the spatial shape of the polypoid lesion. (b) GM shows not only a polypoid lesion but also contrast coating its surface and layering in its inferior portion. (c) EB exploration of the lesion up to 5 voxels below the surface shows gray-level changes from contrast to soft tissue densities. (d) The correlated 2D position (cross) of the 3D EB image in c (cross). The density indicates enhanced soft tissue (HU, 77).

effects are approved by the clinical evaluation. Consequently, the clinical classification and diagnostic performance are also improved with statistically significance.

CTE has great potential for diagnosis, surgery training, and treatment planning because it has fewer restrictions than does standard endoscopy, as the virtual camera can go anywhere and the procedure is more comfortable for the patient [12].

SR is the routine display method used for virtual CT endoscopy. SR for the screening of endoluminal lesions is rapid and has a very good 3D steric effect. VR is also a common rendering technique, introducing color and transparent functions to simulate a life-like colonic mucosal surface. However, SR and monochromatic VR are not sufficient for visualizing and examining the nature of intraluminal lesions, efficient image correlation, and further differentiation.

Because most radiologists and physicians are familiar with the gray-level changes on ordinary abdominal CT, we opt to use gray level instead of color for visualization. GM/TGM

**Table 1. Image features and classification scores of endoluminal lesions on CTC.**

| Lesion Class | Case number | Rated feature | SR | GM | TGM | EB | P * vs. SR | Agreement EB vs. SR |
|---|---|---|---|---|---|---|---|---|
| **Potential malignancy** | 34 | Spatial shape | 2 | 2.44 | 3.00 | 3 | <0.01 | 33(97%) |
| | | | 2 | 2.97 | 2.97 | 2.94 | | |
| | | texture | 1 | 2.03 | 2.44 | 2.98 | <0.01 | 34(100%) |
| | | | 1 | 2.29 | 2.32 | 2.92 | | |
| | | Correlation | 1.9 | 2.47 | 2.94 | 3 | <0.01 | 33(97%) |
| | | | 1.92 | 2.94 | 2.94 | 2.94 | | |
| **Benign** | 28 | Spatial shape | 2 | 2.46 | 2.96 | 3 | <0.01 | 26(93%) |
| | | | 2 | 2.93 | 2.93 | 2.88 | | |
| | | texture | 1 | 2.07 | 2.5 | 2.99 | <0.01 | 28(100%) |
| | | | 1 | 2.32 | 2.43 | 2.73 | | |
| | | Correlation | 1.31 | 2.00 | 2.67 | 2.41 | <0.01 | 27(96%) |
| | | | 1.88 | 2.54 | 2.64 | 2.76 | | |

Potential malignancy: polyps or tumors

Benign lesions: stool, fluid feces, ileocecal valves, intraluminal calcification

* Wilcoxon signed ranks test (one-tailed)

Upper and lower sub-rows of every rated parameter indicate the average score of two different radiologists.

The values of power of all paired tests (EB vs. SR) are above 0.8

applies the gray level to the colon surface using the gray level of the voxel inside the colon wall that is nearest to the surface point. Readers can interpret these gray-level changes in the 3D scene as they do in 2D images. GM/TGM has the further unique advantage of allowing precise 2D to 3D point-to-point correlation of location and gray level. However, the evaluations are only on the surface and can not reach the inside the lesion for further differentiation.

During optic endoscopy, an identified endoluminal lesion is often biopsied. The same concept can be applied to CTC. Akin to the removal of superficial soft tissue during actual endoscopy, the surface points of an endoluminal lesion in CTC can be removed layer by layer to show the gray-level pattern inside the lesion. This EB procedure in CTC is useful for the evaluation of lesions inside the surface.

The tools we developed in this study allowed us to evaluate any ROI during the navigation. The design of ROI gray-level mapping lets the user focus on the target lesion for further

**Table 2. Diagnostic performance of SR, GM, TGM and EB.**

| Reviewer/ display method | Sensitivity (95% CI) | Specificity (95% CI) | Diagnostic accuracy | P * |
|---|---|---|---|---|
| **Reader 1** | | | | |
| SR | 97.1% (0.851; 0.995) | 17.9% (0.079; 0.356) | 61.3% | 0.082 |
| GM | 100% (0.899,1) | 64.3% (0.458,0.793) | 83.9% | <0.001 |
| TGM | 100% (0.899,1) | 85.7% (0.685,0.943) | 93.5% | <0.001 |
| EB | 100% (0.899; 1) | 92.9% (0.774; 0.980) | 96.8% | <0.001 |
| **Reader 2** | | | | |
| SR | 88.2% (0.734; 0.953) | 14.3% (0.057; 0.315) | 54.8% | 1.000 |
| GM | 97.1% (0.851,0.995) | 53.6% (0.358,0.705) | 77.4% | <0.001 |
| TGM | 100% (0.899,1) | 75.0% (0.566,0.873) | 88.7% | <0.001 |
| EB | 100% (0.899; 1) | 89.3% (0.728; 0.963) | 95.2% | <0.001 |

CI, confidence interval

*Values represent comparison between final pathology and diagnostic performance of SR and EB

exploration, without interfere from gray-level changes in the background. This method also reduces the computation time, because only the small ROI area is processed and most background area remains unchanged.

CTC imaging technology enables both 2D and 3D evaluation of endoluminal lesions. However, errors resulting from a single 2D interpretation [13, 14] and primary 3D evaluation [15] are reported. To increased sensitivity and specificity of CTC, All current consensus guidelines acknowledge the complementary roles of 2D and 3D visualization methods, and point-to-point correlation between the two is favored [16]. Although most current CTC analysis software programs support simultaneous display of the cursor position in the 3D and multi-planar 2D images, by placement of it in either the 2D or 3D image. However, the voxel value is difficult to differentiate and it is hard to do 3D and 2D image correlation. CTC readers must go back and forth from 3D to 2D, or 2D to 3D for image correlation to achieve adequate diagnosis. GM/TGM facilitates this condition and enables gray level presentation of the 3D CTC scene as a whole at voxel base to represent air, calcifications, contrast, enhanced tumors, fat, non-enhancement feces and colonic wall either in the initial or following biopsy surface during the process. A different voxel location reflects gray level value of the different corresponding point in the 2D image. EB extends the GM/TGM technique to the deep layers inside the lesion. Therefore, the complete point-to-point correlation between the 2D and 3D images by the location and gray level value can be achieved.

As follows, we summarize and illustrate the novel findings of several target endoluminal lesions those have not been observed in traditional 3D CTC image. 1) Stool: A shaggy appearance and heterogeneous lower surface densities resulting from the lack of contrast enhancement and inner non-GM pore due to air cavities are seen in EB images (Fig 3C) but not in SR images. The correlation between the 2D position and gray-level value of the hypo-dense areas and air bubbles in EB image (Fig 3C arrow and cross) is shown in Fig 3D (arrow and cross). (HU of hypo-density, −110). 2) Tumor/polyp: Increased densities of the interior of the tumor (Fig 4B) due to contrast enhancement are seen in EB images but not in SR images (Fig 4A). The correlated interior position and higher gray-level value in the 2D image are shown in Fig 4B and 4C cross. (HU, 57). 3) Fat: The presence of fat inside the protruding luminal area can be observed in EB images (Fig 5C). The correlated 3D and 2D positions of fat content are shown in Fig 5C and 5D (HU, −88). 4) Calcifications or tagging agents: These features are only observed in the GM images (Fig 6B) and cannot be differentiated in the SR image (Fig 6A). The EB method is superior to SR for visualizing lesions beneath the tagging agents (Fig 6C). The correlated 3D and 2D positions of the polypoid lesion beneath the tagging agents are shown in Fig 6C and 6D (cross) (HU, 77).

In clinical evaluations, EB was statistically significantly better than traditional SR display in the 3D shape and gray-level pattern effects both on the surface and inside of endoluminal lesions in CTC (P < 0.01) with high power and interobserver agreement. The EB effects are comparable with GM/TGM method but significantly increase more gray level presentation than GM/TGM (P<0.01). The sensitivity, specificity, and accuracy of EB was higher than those of SR in solitary 3D CTC readings. The specificity and diagnostic accuracy will be increased gradually following the order of GM applied to CTC, TGM of the endoluminal lesion and EB for internal exploration. The result implies the focus and internal gray level evaluation is helpful for differentiation of a lesion. In addition, the user can correlate its 2D images not only by spatial shape but also by gray level information in EB, as performed by GM/TGM but no in SR CTC. However, there is no more surface restriction in GM/TGM. The user can recheck any 3D point in EB, either inside or on the colon surface in 2D images, according to the location and gray level value to facilitate accurate diagnosis.

Translucency mapping has been used to differentiate polyps from peudopolyps and obtained excellent results [17]. The EB is quite different from the translucency rendering

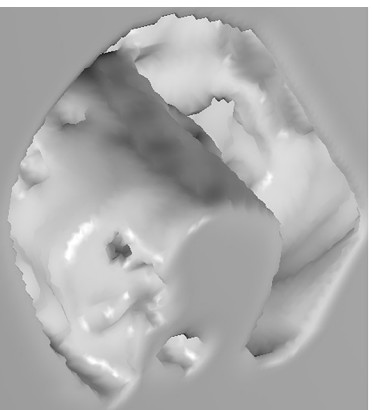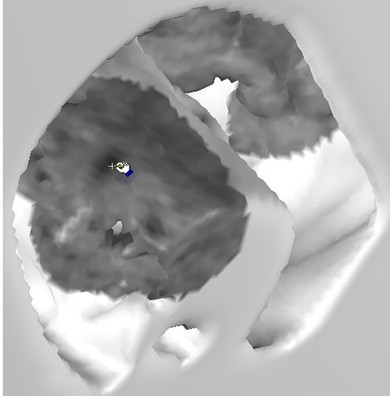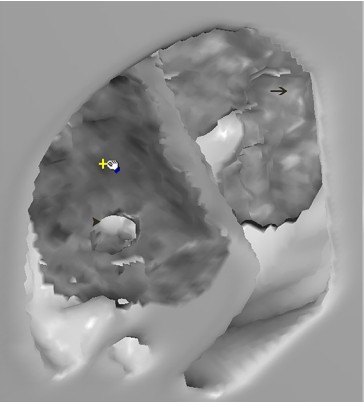

**Fig 7. Noncontrast CTC Images of a 74-year-old female with a descending colon cancer.** (a) An SR image of an annular colon mass and a proximal impacted polypoid lesion. (b) An initial EB image shows a mass with higher density and smooth surface (arrow). Another polypoid lesion with heterogeneous lower densities (cross), shaggy and porous appearance proximal to the above mass. (c) An EB image up to 4–5 voxel depth to the surface shows the persistent and apparent above features (arrow and cross) corresponding to those of (b). The cavitation of the polypoid lesion becomes more apparent (arrow head) compared with that of (b). An annular colon cancer and proximal impacted stool were proved by the colonoscopy. The non-contrast EB features are similar to the contrast Figs 3(C) and 4(C) images.

where all the grey level values of the voxels along a ray are incrementally summed up by an alpha blending equation [18]. Working as a computer-aided diagnosis tool, translucent rendering is subjective and influenced by user settings, such as the assignment of blue, green, red, and white color channels to areas of increasing attenuation and the number of voxels involved in the translucent computation along a ray. The real depth, size, gray level pattern, 2D and 3D image correlation of the potential lesion are still unknown. The EB can be worked together with the translucent rendering for further investigation of the compositions and fine judgement inside a potential lesion. In other words, the translucent rendering algorithm can be used for the further analysis and confirmation of polyps after the EB examination.

To test the EB effect in the non-contrast images, one of the cases with proven colon cancer, and impacted stool without intravenous contrast administration, was selected to reconstruct the 3D images and demonstrate the effects. These non-contrast CTC images (Fig 7) are compared with the contrast CTC images in Figs 3 and 4. Tumors show consistent higher soft tissue density with smooth surface (Fig 7B, arrow), whereas stool shows consistent heterogeneous lower densities (Fig 7B, cross) with shaggy and porous appearance either in the non-contrast or contrast initial EB images (Figs 4B and 7B). Those features and characteristic stool cavitations are persistent and become more apparent in the subsequent layer by layer EB images (Figs 4C and 7C). Generally colonic tumor, stool, fluid, fat and tagging agents are significantly attenuation different in non-contrast CT images and sufficient in screening CTC. The differences are reflected directly and natively in the GM EB images. However, intravenous contrast administration will enhanced the attenuation difference and achieve a better diagnostic results.

This study has limitations. A retrospective study of a small number of pre-selected images in this preliminary study. However, the very small p ($<0.01$) and high power ($>0.8$) values warrant the adequate sample size to demonstrate the advantage of EB technique. Special cases like too small polyps, flat lesions, the polyps completely submerged in tagged fecal materials/fluid or extraluminal lesions that are invisible in 3D views are not discussed in the study. Artifacts such as beam hardening due to adjacent fecal-tagged materials or arthroplasty cause failures of CTC and following EB. Shading may interfere with subtle gray-level changes on the

surface and inside endoluminal lesions. Color mapping of potential malignancies to further facilitate the discrimination of lesions warrants future study. Translucent mapping is definitely an important reference for the study. Combination of it with EB is a potential way to further predict, analyze and localization of the malignancies.

In summary, we have developed tools to integrate gray level value (HU) and 3D steric shape information layer by layer into endoluminal lesion images. The tools also display the corresponding location and gray level value in 2D images on and inside the lesion surface. Clinical experiments show that EB is significantly better than SR in all respects, including spatial shape, gray-level patterns inside or on the surface, and 2D image correlation and classification. Its gray level presentation is significantly better than that of GM or TGM with the similar spatial and correlation effects. EB increases the accuracy of identification and differentiation of benign vs. malignant endoluminal lesions over that of SR, GM or TGM. EB serves as a suitable adjunct tool for further evaluation in the endoluminal view. We anticipate that our technique will be applicable to other air-filled tubular structures, such as digestive and airway systems, to facilitate clinical diagnosis and management.

## Author Contributions

**Conceptualization:** Lih-Shyang Chen, Shao-Jer Chen, Chun-Ju Hou, Ku-Yaw Chang.

**Data curation:** Lih-Shyang Chen, Shao-Jer Chen, Ta-Wen Hsu, Shu-Han Chang, Chih-Wen Lin, Yu-Ruei Chen, Chin-Chiang Hsieh, Shu-Chen Han.

**Formal analysis:** Lih-Shyang Chen, Shao-Jer Chen, Ta-Wen Hsu, Shu-Han Chang, Chun-Ju Hou, Chih-Wen Lin, Yu-Ruei Chen, Chin-Chiang Hsieh, Shu-Chen Han.

**Funding acquisition:** Lih-Shyang Chen, Shao-Jer Chen, Ku-Yaw Chang.

**Investigation:** Lih-Shyang Chen, Shao-Jer Chen, Shu-Han Chang, Chih-Wen Lin, Ku-Yaw Chang.

**Methodology:** Lih-Shyang Chen, Shao-Jer Chen, Shu-Han Chang, Chun-Ju Hou, Ku-Yaw Chang.

**Project administration:** Lih-Shyang Chen, Shao-Jer Chen, Ta-Wen Hsu, Shu-Han Chang, Chih-Wen Lin, Yu-Ruei Chen, Chin-Chiang Hsieh, Shu-Chen Han.

**Resources:** Lih-Shyang Chen, Shao-Jer Chen, Ta-Wen Hsu.

**Software:** Lih-Shyang Chen, Shu-Han Chang.

**Supervision:** Lih-Shyang Chen, Shao-Jer Chen, Ta-Wen Hsu, Chun-Ju Hou, Ku-Yaw Chang.

**Validation:** Lih-Shyang Chen, Shao-Jer Chen, Ta-Wen Hsu, Shu-Han Chang, Chun-Ju Hou, Chih-Wen Lin, Yu-Ruei Chen, Chin-Chiang Hsieh, Shu-Chen Han.

**Visualization:** Lih-Shyang Chen, Shao-Jer Chen, Shu-Han Chang, Chih-Wen Lin, Yu-Ruei Chen, Chin-Chiang Hsieh, Shu-Chen Han, Ku-Yaw Chang.

**Writing – original draft:** Shao-Jer Chen.

**Writing – review & editing:** Lih-Shyang Chen.

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
