## [Decision Letter · Decision Letter 0]

9 Jun 2022

PONE-D-21-31120

Exploring the interior of 3D endoluminal lesions in the air spaces by a novel electronic biopsy technique: a preliminary study of endoluminal colon lesions.

PLOS ONE

Dear Dr. Chen,

Thank you for submitting your manuscript to PLOS ONE. After careful consideration, we have decided that your manuscript does not meet our criteria for publication and must therefore be rejected.

The external reviewer and I have now evaluated the manuscript. There are major points that limited a positive decision to be drawn on the manuscript. You could find the comments below. In particular please pay attention to the methodology and design:

The small sample size without power calculation; The retrospective design of the study.

I am sorry that we cannot be more positive on this occasion, but hope that you appreciate the reasons for this decision.

Kind regards,

Zubing Mei, MD,PH.D

Academic Editor

PLOS ONE

Reviewers' comments:

Reviewer's Responses to Questions

**Comments to the Author**

1. Is the manuscript technically sound, and do the data support the conclusions?

Reviewer #1: Yes

2. Has the statistical analysis been performed appropriately and rigorously? 

Reviewer #1: Yes

3. Have the authors made all data underlying the findings in their manuscript fully available?

Reviewer #1: Yes

4. Is the manuscript presented in an intelligible fashion and written in standard English?

Reviewer #1: Yes

5. Review Comments to the Author

Reviewer #1: Review Report

• In this paper, the authors aimed to investigate the application of an ‘electronic biopsy’ (EB) technique to computed tomographic colonography (CTC).

• Based on a retrospective analysis of 30 patients with 62 endoluminal lesions of various types, the authors conclude

• The paper is interesting, well structured, and correctly organized. The authors have clearly worked hard to detail their study, but I have some comments:

POINTS OF WEAKNESS

1. Retrospective design.

2. Small sample size.

SPECIFIC COMMENTS

1. The type of study and the number of included patients should be mentioned in the abstract.

2. A flow chart of the study is needed.

3. What was the power of sample size calculation?

4. More details about CT parameters are needed

5. Why did the authors not perform interviewer agreement to strengthen their results?

6. PLOS authors have the option to publish the peer review history of their article (what does this mean?). If published, this will include your full peer review and any attached files.

Reviewer #1: **Yes: **Mohammad Abd Alkhalik Basha

- - - - -

---

## [Author Response · Author response to Decision Letter 0]

19 Feb 2023

First of all, we would like to thank the anonymous reviewer for giving the constructive suggestions to improve this manuscript. The responses for the comments are summarized in the following.

Reviewer #1: Review Report

• In this paper, the authors aimed to investigate the application of an ‘electronic biopsy’ (EB) technique to computed tomographic colonography (CTC).

• Based on a retrospective analysis of 30 patients with 62 endoluminal lesions of various types, the authors conclude

• The paper is interesting, well structured, and correctly organized. The authors have clearly worked hard to detail their study, but I have some comments:

POINTS OF WEAKNESS

1. Retrospective design.

Response:

 Page 24, the second paragraph: This study has limitations. A retrospective study of a small number of pre-selected images in this preliminary study. However, the very small p (<0.01) and high power (>0.8) values warrant the adequate sample size to demonstrate the advantage of EB technique.

2. Small sample size.

Response:

 Page 24, the second paragraph: This study has limitations. A retrospective study of a small number of pre-selected images in this preliminary study. However, the very small p (<0.01) and high power (>0.8) values warrant the adequate sample size to demonstrate the advantage of EB technique.

SPECIFIC COMMENTS

1. The type of study and the number of included patients should be mentioned in the abstract.

Response:

 Abstract, line 4: A retrospective study of sixty-two various endoluminal lesions from thirty patients (13 males, 17 females; age range, 31 to 90 years) was approved by our institutional review board and evaluated.

2. A flow chart of the study is needed.

Response:

 Page 7, the first paragraph: The flow chart methodology of this research is shown in the figure 1. 

3. What was the power of sample size calculation?

Response:

 Page 15, the second paragraph: EB scored significantly higher than did SR in the criteria evaluated by the two radiologists, regardless of the malignant or benign status as determined pathologically (one-tailed Wilcoxon signed ranks test, P < 0.01). The power values of the paired tests are also high (> 0.8). According to the localization and gray level correspondence, EB images provided a significantly higher correlation of 3D to 2D images than did SR images (one-tailed Wilcoxon signed ranks test, P < 0.01) with high power above 0.8 (Table 1).

4. More details about CT parameters are needed

Response:

 Page 7, the second paragraph: Helical CT scans of the prone and supine positions were performed 30 seconds after intravenous contrast administration with the following parameters: collimation 64×0.6 mm; gantry rotation time 0.8 s; X-ray Tube Current: 140 mA; Exposure Time: 800 ms; kVp 120. Scanning was performed craniocaudally during a single breath-hold. With the advance of multiple-slice CT, the slice thickness of volume data can be reconstructed to 1.25 mm. All scans were performed using a 64-slice GE Lightspeed VCT scanner (GE HealthCare, Milwaukee, Wisconsin, USA).

5. Why did the authors not perform interviewer agreement to strengthen their results?

Response:

 Page 15, the second paragraph: The classifications performed by both observers were in very good agreement—that is, better EB experience than SR (Table 1).

---

## [Decision Letter · Decision Letter 1]

27 Mar 2023

Exploring the interior of 3D endoluminal lesions in the air spaces by a novel electronic biopsy technique: a preliminary study of endoluminal colon lesions.

PONE-D-21-31120R1

Dear Dr. Chen,

We’re pleased to inform you that your manuscript has been judged scientifically suitable for publication and will be formally accepted for publication once it meets all outstanding technical requirements.

Kind regards,

Shuai Ren

Academic Editor

PLOS ONE

Additional Editor Comments (optional):

Congratulations on the good work! The paper can be accepted in its current form.

Reviewers' comments:

Reviewer's Responses to Questions

**Comments to the Author**

1. If the authors have adequately addressed your comments raised in a previous round of review and you feel that this manuscript is now acceptable for publication, you may indicate that here to bypass the “Comments to the Author” section, enter your conflict of interest statement in the “Confidential to Editor” section, and submit your "Accept" recommendation.

Reviewer #1: All comments have been addressed

2. Is the manuscript technically sound, and do the data support the conclusions?

Reviewer #1: Yes

3. Has the statistical analysis been performed appropriately and rigorously? 

Reviewer #1: Yes

4. Have the authors made all data underlying the findings in their manuscript fully available?

Reviewer #1: Yes

5. Is the manuscript presented in an intelligible fashion and written in standard English?

Reviewer #1: Yes

6. Review Comments to the Author

Reviewer #1: The authors have performed a good job and responded to all reviewers comments

The paper may be accepted in its current form

7. PLOS authors have the option to publish the peer review history of their article (what does this mean?). If published, this will include your full peer review and any attached files.

Reviewer #1: No

---

## [Editor Report · Acceptance letter]

4 May 2023

PONE-D-21-31120R1 

Exploring the interior of 3D endoluminal lesions in the air spaces by a novel electronic biopsy technique: a preliminary study of endoluminal colon lesions. 

Dear Dr. Chen:

I'm pleased to inform you that your manuscript has been deemed suitable for publication in PLOS ONE. Congratulations! Your manuscript is now with our production department. 

Kind regards, 

on behalf of

Dr. Shuai Ren 

Academic Editor

PLOS ONE